# Proximal Femoral Fractures in the Elderly: A Few Things to Know, and Some to Forget

**DOI:** 10.3390/medicina58101314

**Published:** 2022-09-20

**Authors:** Nicola Maffulli, Rocco Aicale

**Affiliations:** 1Department of Musculoskeletal Disorders, Faculty of Medicine and Surgery, University of Salerno, 84084 Baronissi, Italy; 2Clinica Ortopedica, Ospedale San Giovanni di Dio e Ruggi D’Aragona, 84131 Salerno, Italy; 3Barts and the London School of Medicine and Dentistry, Queen Mary University of London, Centre for Sports and Exercise Medicine, Mile End Hospital, London E1 4DG, UK; 4School of Pharmacology and Bioengineering, Guy Hilton Research Centre, Faculty of Medicine, Keele University, Thornburrow Drive, Hartshill, Stoke-on-Trent ST4 7QB, UK

**Keywords:** elderly, nailing, hip, proximal hip fracture, COVID-19

## Abstract

Hip fractures are a leading cause of hospitalisation in elderly patients, representing an increasing socioeconomic problem arising from demographic changes, considering the increased number of elderly people in our countries. Adequate peri-operative treatment is essential to decrease mortality rates and avoid complications. Modern management should involve a coordinated multidisciplinary approach, early surgery, pain treatment, balanced fluid therapy, and prevention of delirium, to improve patients’ functional and clinical outcomes. The operative treatment for intertrochanteric and subtrochanteric fractures is intramedullary nail or sliding/dynamic hip screw (DHS) on the basis of the morphology of the fracture. In the case of neck fractures, total hip replacement (THR) or hemiarthroplasty are recommended. However, several topics remain debated, such as the optimum thromboprophylaxis to reduce venous thromboembolism or the use of bone cement. Postoperatively, patients can benefit from early mobilisation and geriatric multidisciplinary care. However, during the COVID-19 pandemic, a prolonged time to operation with a subsequent increased complication rate have burdened frail and elderly patients with hip fractures. Future studies are needed with the aim to investigate better strategies to improve nutrition, postoperative mobility, to clarify the role of home-based rehabilitation, and to identify the ideal analgesic treatment and adequate tools in case of patients with cognitive impairment.

## 1. Introduction

Proximal femur fractures are a common consequence of osteoporosis, and we refer collectively to them as “hip fractures”. They are a global challenge for healthcare systems and for patients themselves and their families, as there were 1.31 million of hip fractures in 1990 [1], and they are predicted to rise to 6.26 million globally by 2050 [2,3]. The socio-economic costs represent 0.1% of the global burden of disease worldwide [1]. Hip fractures are potentially a catastrophic event: about 30% of such patients will die within the first year after injury [4], and the survivors will experience an increasing ongoing burden of illness which will affect their quality of life [5]. Within 1 year following the fracture, only between 40 and 60% of such elderly patients will have returned to their pre-injury level of mobility and ability [6].

Several evidence-based guidelines are supported by systematic reviews, and such patients commonly present the association of different metabolic (diabetic and thyroid disease) and inflammatory diseases [7,8,9,10,11].

Patients older than 80 or patients with common elderly multimorbidity aged over 70 are defined as “geriatric” [12], and 25 to 50% of patients over 85 are considered frail [13]. Frailty is a specific condition described as an increased vulnerability to stressors [13], and frailty fractures are defined as bone damage in the absence of important trauma or following a fall from standing height or less; in this context, hip fractures are the most common type of frailty fracture [14,15].

Surgical management should take place within the first 24 h, beyond which there is an increased chance of peri-operative complications (i.e., pulmonary embolism, pneumonia, deep vein thrombosis (DVT), urinary tract infections). In case of surgery delay for more than 48 h, mortality may rise significantly [16]; however, if surgery is undertaken within 48 h, a 20% lower risk of dying during the next year has been reported [17].

During the COVID-19 global pandemic, the total number of hip fracture patients was significantly reduced [18]. However, a systematic review and meta-analysis have shown a seven-fold increased mortality risk for COVID-19-positive patients with hip fractures and an increase in postoperative complications [19]. The time necessary to obtain the results of COVID-19 tests, the reduced operating capacity, and the shortage of hospital staff were identified as the major challenges. A recent multicentre study showed a mean delay of 2.4 days to surgery, with a minimum of 0 days and a maximum of 13 days [20].

The identification and treatment of geriatric conditions and prevention of complications is the aim of a comprehensive geriatric assessment [7], and modern “hip fracture care” is a multidisciplinary effort which acknowledges that a hip fracture is not a simple fracture but a marker of general health status deterioration.

## 2. Diagnosis

Typically, proximal femoral fractures occur in the elderly as a result of low energy trauma (i.e., a fall from standing). In the UK, the last report of the National Hip Fracture database (NHFD) reveals that 91.6% of hip fractures occur in patients over 70, and 72% are females [21], reflecting the increasing probability of falling (in the over 65 years, one in three people fall each year) and osteoporosis with advancing age [22].

On examination, patients report hip pain and inability to bear weight, with the affected leg shortened and externally rotated. Plain radiographs are adequate for diagnosis, but, when they are apparently normal with clinical signs and symptoms suggestive of a hip fracture, magnetic resonance imaging (MRI) or computed tomography (CT) may be indicated, i.e., the so-called ”occult hip fracture”, [23].

## 3. Classification

Hip fractures can be divided into intra and extracapsular, respectively, inside or outside the hip joint capsule, reflecting the disrupted blood supply of the femoral head, and guiding the decision process as to whether the patient will undergo (hemi) arthroplasty or internal fixation of intra-capsular fractures, and the choice of which construct to use to stabilise extracapsular fractures (i.e., intramedullary fixation with a nail or extramedullary fixation with a sliding hip screw) (Figure 1) [2,24].

Generally, patients will undergo surgery, obtaining benefits of the early fixation/replacement such as rapid postoperative mobilisation, and avoiding the poor outcomes and risks associated with long-term immobilisation from nonoperative treatment [25].

Intracapsular fractures are commonly divided in subcapital, midcervical, and basicervical; especially in the elderly, midcervical are the most common type, at over 86% of intracapsular fractures [26].

Three classifications for femoral neck fractures are the most common used: Garden’s, Pauwels’s (Figure 2 and Figure 3), and the AO classification.

The Garden classification, characterised by a fair inter-observer reliability, is composed of four types: type I describes an incomplete or impacted fracture; type II a complete fracture without displacement; type III a complete fracture with partial displacement; and type IV a complete fracture with full displacement [27].

The Pauwels classification is based on biomechanical forces and pressure at the fracture line site: in type I, a compression force is dominating, with a fracture line up to 30° to the horizontal plane; in type II there is a shearing stress, with negative impact on bone healing [28] and with a fracture line between 30° and 50°; in type III, the fracture line is above 50° with shearing stress being the predominant force, leading to fracture displacement [29]. However, for this classification, a weak reliability and reproducibility have been reported [30].

Probably, the most complete classification is the AO classification which combines the fracture level, degree of displacement, and fracture line angle. The original version of the (AO)/ASIF classification for hip fracture has been in use since 1990 [31], and it has rapidly become popular and readily used in the scientific literature. The new AO/OTA classification, published in 2018, imparts greater importance to the integrity of the lateral wall, which may play an important role in decision making and has been identified as a major prognostic factor to predict mechanical failure after surgery [32]. Furthermore, the AO/OTA classification considers isolated trochanteric fractures (of the greater or lesser trochanter), which were not classified in the original AO system.

## 4. Peri-Operative Pharmacological Management

Pain management is mandatory given its essential role in delirium prevention [33]. However, in the elderly, NSAIDs are not recommended, and drugs such as paracetamol every 6 h, unless contraindicated, can be useful [34]. When pain control is not achieved, oral opioids can be administered and accompanied by constipation prophylaxis [16].

Routine laboratory tests should include complete blood count, inflammation markers, prothrombin time—international normalised ratio (PT-INR), partial thromboplastin time, and metabolic profile [16]. Given their age, frequently, patients with hip fractures tend to be dehydrated; a flow rate of 100–200 mL/h of isotonic crystalloids is estimated to be safe [16].

The incidence of urinary tract infections and asymptomatic bacteriuria increases with age [35], and an association with superficial wound infections and symptomatic bacteriuria has been reported. A recent systematic literature review has shown that the postoperative infectious rate did not decrease if asymptomatic bacteriuria was treated before surgery [36]. Therefore, screening of the urinary tract infections is recommended, but treatment is needed only when symptomatic [16].

Thromboprophylaxis received great attention for hip fracture patients in the last few years given the risk of deep vein thrombosis (DVT), but the role of early surgery and mobilisation in mitigating this risk is clear.

Given the potential increase in morbidity and mortality from thromboembolic events, several national guidelines recommend thromboprophylaxis [8] and, at present, some evidence can be found supporting the graduated compression stockings and cyclical leg compression devices to reduce DVT with relatively good compliance and little risk of skin abrasions [37,38].

Regarding bleeding complications prevention, 40% of elderly patients with hip fractures take anticoagulants or antiplatelet agents [39], and optimal coordination with anaesthesiologists is mandatory: in patients with antiplatelet therapy, the recommendation is to proceed with the surgery with no delay [40]. In the case of double antiplatelet therapy, spinal anaesthesia is contraindicated [40]. Furthermore, a PT-INR value below 1.5 is an indication for vitamin K antagonists, including warfarin and phenprocoumon [40].

The use of clopidogrel and aspirin can increase perioperative blood loss, but hip fracture surgery can still safely be performed with no delay [41].

In patients with mechanical valves, atrial fibrillation (AF), with recent stroke history, DVT, or pulmonary embolism, the use of subcutaneous low-molecular weight heparin or intravenous unfractionated heparin need to be taken into consideration [42].

In the case of patients who use anti-Xa-agents (Apixaban, Edoxaban, Rivaroxaban), a plasma drug level of under 50 pg/mL is deemed safe for surgery, and, if the plasma level cannot be measured, a 24 h gap between the last dose and surgery should be considered [43].

Systemic tranexamic acid administration reduces blood loss and transfusion rates, impacting favouirably on post-operative bleeding and not interfering with anti-coagulation. However, a recent meta-analysis could not ascertain what its optimal regimen, timing, and dosage are [44].

Delirium can be present, and often it remains undiagnosed in the elderly [45], increasing complications and mortality risks. Its prevention can play an essential role in the care of hip fracture patients [46]. Screening for delirium is not simple, but questionnaires such as the 4AT, a sensitive and specific tool, are validated for hip fractures [47], and can be used to evaluate mental status changes. They should be used in routine screening on admission. Multicomponent non-pharmacological approaches have been used, showing good results and including early mobilisation, adequate hydration, sleep enhancement, orientation in time and place, and therapeutic activities such as reminiscence [45]. An ideal policy for visitors can be adjusted to achieve a reduction in stress and maintain routine activities and a normal night–day rhythm.

## 5. Surgical Management

Hip fractures are an emergency, and strong evidence regarding early surgery is associated with a reduction in the risk of death [48].

Treatment should aim to return patients to their previous levels of daily life activities and full weight bearing. Management depends on the different type of hip fracture, based on the vascular anatomy of the proximal femur and the different chances of bone healing and future complications.

Regarding intertrochanteric and subtrochanteric fractures, surgical management is intramedullary nailing, which allows the decrease in soft tissue injuries during surgery and early weight-bearing after surgery (Figure 4). The implant choice for intertrochanteric fractures depends on fracture stability defined by the lateral cortical wall [49]. For example, extramedullary devices such as the sliding hip screw (SHS) can be chosen when the lateral cortex is intact, but an intramedullary device has biomechanical advantages given its location closer to the vector of the force of gravity, due to a shorter lever arm compared to extramedullary devices [24,49].

A recent meta-analysis comparing different management options for intertrochanteric fractures [dynamic hip screw, compression hip screw, percutaneous compression plate, Medoff sliding plate, less invasive stabilisation system, gamma nail, proximal femoral nail, and proximal femoral nail anti-rotating (PFNA)] identified the PFNA as the option with less blood loss and higher functional results [50].

The use of the helical blade in intramedullary devices resulted in a higher collapse rate of the neck-shaft angle with a cut-out of the screw compared to the lag screw [51].

A recent prospective randomised controlled trial in hip fracture patients showed that the use of nail and cephalic hydroxyapatite coated screws results in higher mechanical stability and improved implant osteointegration compared to standard nailing [52].

A less common type of hip fracture is the subtrochanteric fracture, for which intramedullary nailing with a long nail is the accepted standard, given the reduced operation time, fixation failure rate, and length of stay (LOS) when compared to extramedullary devices [53].

SHSs are an established and optimal option to manage extra-capsular hip fractures, in particular the extra-capsular AO/OTA A1 and A2 fractures avoiding fracture collapse with good mechanical stability [54,55]. However, in the case of more complex unstable fractures (A3 types) with comminution and/or deficient bone to share the load with the fixation device, the fracture may collapse into varus with the consequent cut-out of the cephalic screw, or the femoral shaft may medialise excessively producing mechanical failure (Figure 5). An intramedullary nail for subtrochanteric fractures, and these types of fractures, achieves a more stable construct [56].

Despite the clear guidelines about the use of modern implants in certain fracture patterns, there still remain some gaps in the evidence [57].

Cement augmentation improves the stability of the implant in osteoporotic bone, but it has been linked to the risk of thermal damage, osteonecrosis, and cement leaking at the fracture site [58]. A recent systematic review on the clinical results of cement augmentation showed improved radiographic parameters and lower complication rates, but more studies are needed [59].

Femoral neck fractures can be managed conservatively or with surgery, using total hip arthroplasty (THA) or hemiarthroplasty. In the case of non-surgical treatment, patients with more than one comorbidity aged above 70 have an 83% risk of secondary dislocations of the fracture [60], making surgery the best choice in elderly patients. Displaced intra-capsular fractures are approximately half of all hip fractures [21], and they occur in a region where the femoral blood supply is tenuous, and healing is unreliable. Hip hemiarthroplasty, in which only the femoral head is replaced, is the treatment of choice, and current evidence supports the use of bone cement [61] (Figure 6).

For the choice of the implant, two main aspects need to be considered: indication for osteosynthesis, and, furthermore, consideration that elderly patients are less compliant to weight-bearing restrictions [62].

Following the Pauwels classification, in type I or II of femoral fractures, internal fixation is indicated. Considering femoral head blood supply, in type III and IV of the Garden classification fracture osteosynthesis is generally not recommended.

Displaced femoral neck fractures are generally accompanied by the disrupted blood supply predisposing to fixation failure; when there are co-existing osteoporosis and age-related bone changes, there is a major increase in the risk of non-unions in the elderly [63].

Osteosynthesis can be suggested as a salvage option or in young patients with non-placed fractures (Figure 7). If patients are bed-bound, surgery is indicated for pain management.

In the case of healthy and active patients, biological age can guide the implant choice: in the case of high functional requirements and lower biological age, indications shift towards THA instead of hemiarthroplasty, which is indicated in the healthy elderly [64] (Figure 8).

Cemented implants are characterised by less postoperative pain and better mobility [61], with better fixation in the osteoporotic bone [65]. However, bone cement has risks, especially in frail patients, with an increased morbidity and mortality in intra and post-operative periods [66]. However, bone cement implantation syndrome is rare, and evidence highlights the reduction in pain and increased functional outcomes compared to uncemented implants [61].

The periprosthetic femoral fracture risk is two times higher in patients above 60 years with uncemented stems compared to cemented stems [67]. For those with an elevated risk and suitable bone quality, to reduce the risks of cement implantation syndrome during surgery, a non-cemented femoral component is indicated.

Cemented THA should be considered for patients with high levels of pre-injury activity and able to walk independently, with no cognitive impairment and medically fit to undergo a longer operation [57].

THA can be associated with a higher dislocation rate [65], but, in young and active patients, it remains the implant of choice given the optimal outcomes and lower long-term reoperation rate compared to hemiarthroplasty. The risk of dislocation is related to the components’ positioning, surgeon’s experience, and soft tissue tension, [68]. In elderly patients, sarcopenia, proprioception loss, and increased risk of falls are other factors which need to be considered [68] (Figure 9).

Hemiarthroplasty does have some advantages, such as shorter surgery time and lower dislocation incidence [64], but, in young patients, hemiarthroplasty has a high rate of acetabular erosion with the need for conversion in THA for secondary osteoarthritis [69].

A multicentre randomised controlled trial compared displaced femoral neck fractures managed either with THA or hemiarthroplasty, with no difference incidence of secondary interventions, but the better WOMAC score favoured THA over hemiarthroplasty [70].

Basicervical femoral neck fractures are uncommon (1.8% of cases), and management includes both a cephalomedullary nail and DHS. Cancellous screws are not recommended given their high failure rate. Further research with well-defined management outcomes or fixation failure evaluation are needed to achieve clear recommendations [71].

## 6. Postoperative Treatment

To reduce the risk of pneumonia, pressure ulcers, thromboembolism, and delirium, early mobilisation is recommended, particularly in elderly patients [72]. In general, patients who have had one fracture are at risk of another one, and for this reason it is essential to investigate the cause of the fractures and prevent further accidents, taking in consideration that syncope, Parkinson’s disease, and polypharmacy are associated with an increased risk of falling in the elderly [73].

Postoperative care needs to include mechanical thromboembolism prophylaxis mediated by early mobilisation, pharmacological prophylaxis with low molecular-weight heparin continued for 28–35 days, and physiotherapy [74].

## 7. Postoperative Care

Fracture prevention plays an essential role for elderly care, and two strategies are employed: reduce fall risk and improve patients’ overall bone health. To avoid the risk of falls, a clinical assessment to identify medical conditions (such as postural hypotension, syncope, arrhythmia) needs to be undertaken, and basic investigations (i.e., blood pressure measurements, a 12-lead ECG, and a review of current medications) can be helpful. Mechanical causes such as poor mobility and impaired vision need to be evaluated and managed, and a home assessment with relative modifications is recommended.

Bone health status can be obtained by routine blood tests to evaluate calcium or vitamin D deficiency, and a review of drugs used and comorbidities such as liver and renal disease. Secondary prevention of osteoporotic fracture is recommended in elderly patients with confirmed osteoporosis and high risk of re-fracture, with the initial use of anabolic drugs (such as teriparatide, abaloparatide, romosozumab) followed by anticatabolic drugs (i.e., oral or intravenous bisphosphonates or denosumab) [75].

The rehabilitation process begins with the involvement of specialists such as orthogeriatricians, who play a clear role to optimise the patient’s medical condition in the peri-operative period and early supported discharge [39,76]. Mobilisation is recommended already the day after surgery [77], and early intensive rehabilitation is more effective to improve mobility compared to a more sedate approach [78,79].

However, there is no consensus regarding which is the optimal strategy to improve mobility [80]. Only some high-quality studies investigate nutrition’s role [81,82], and moderate evidence [83] supports dietary supplementation, to avoid protein and energy malnutrition, improving postoperative nutritional status and decreasing mortality [81].

## 8. Conclusions

Hip fractures are demanding challenges for patients and healthcare systems. Management cannot be limited to the operating theatre. Given the increase in the burden of disease, the true challenge is in prevention and in developing strategies to improve the quality of life for this group of patients.

Generally, an interdisciplinary orthogeriatric treatment reduces the length of hospital stay, number of complications, and mortality. Essential peri-operative aspects are pain management, early mobilisation, management of fluid, and delirium prevention.

The COVID-19 pandemic has brought additional difficulties in hip fracture patients’ care, leading to a delay in surgery, and a higher complication rate. Despite the importance of this condition and its impact on the life quality of patients, our knowledge is still evolving and there remains a lack of quality evidence for management options that we can offer.

## Figures and Tables

**Figure 1 medicina-58-01314-f001:**
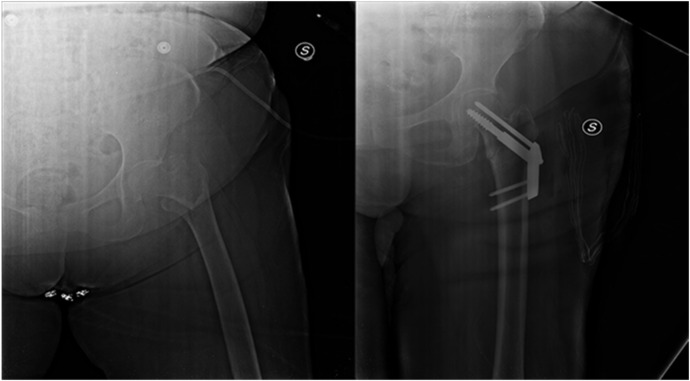
Dynamic hip screw (DHS), pre and post operation of hip fracture using a particular type of DHS, named Anteversa Plate.

**Figure 2 medicina-58-01314-f002:**
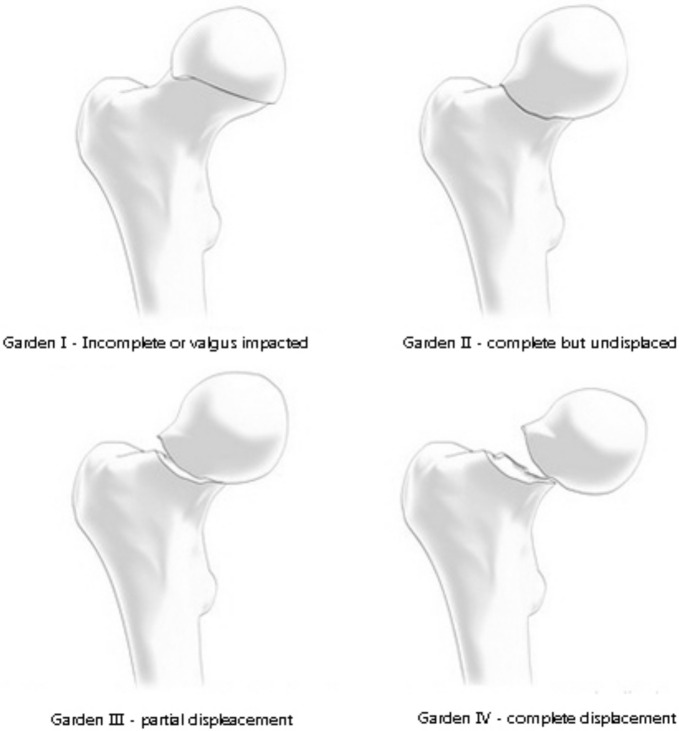
Garden classification.

**Figure 3 medicina-58-01314-f003:**
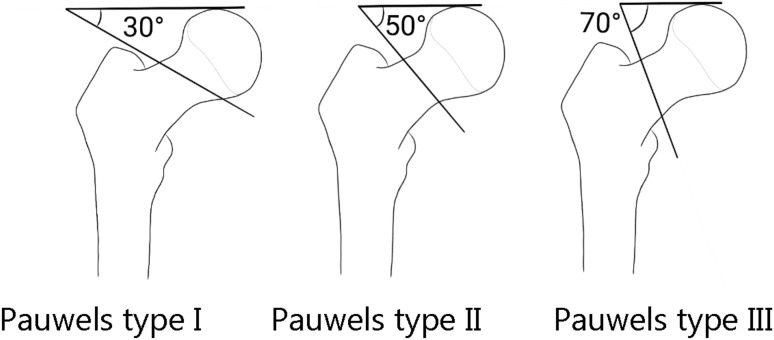
Pauwels classification.

**Figure 4 medicina-58-01314-f004:**
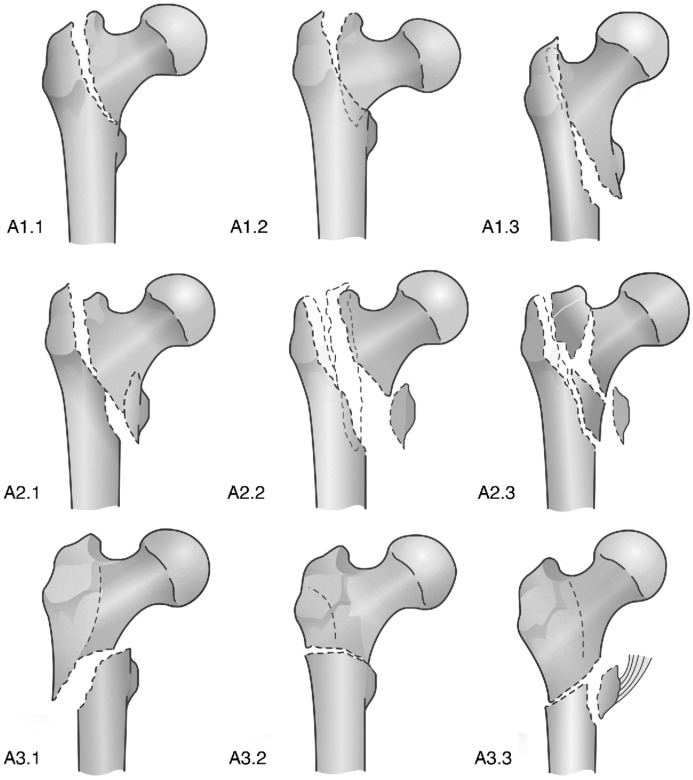
AO/OTA classification and subclassification.

**Figure 5 medicina-58-01314-f005:**
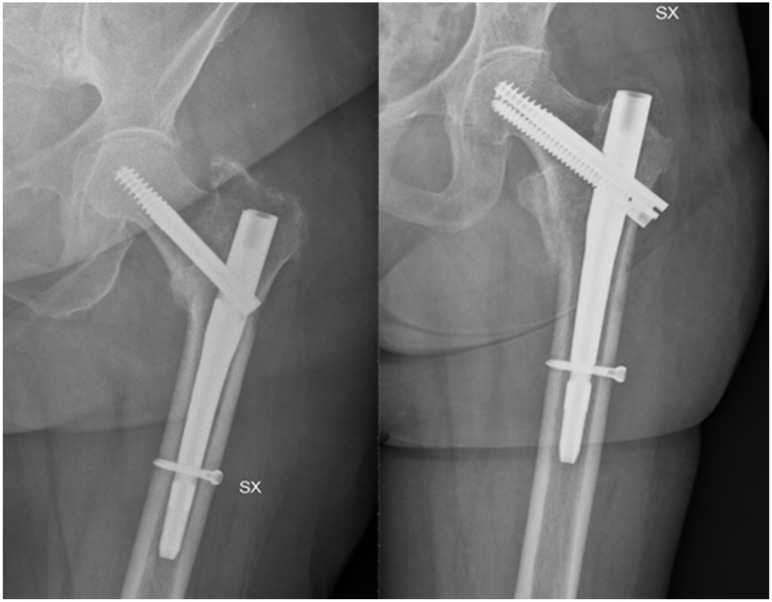
Hip fracture nailing using two different devices characterised by one or two cephalic screws.

**Figure 6 medicina-58-01314-f006:**
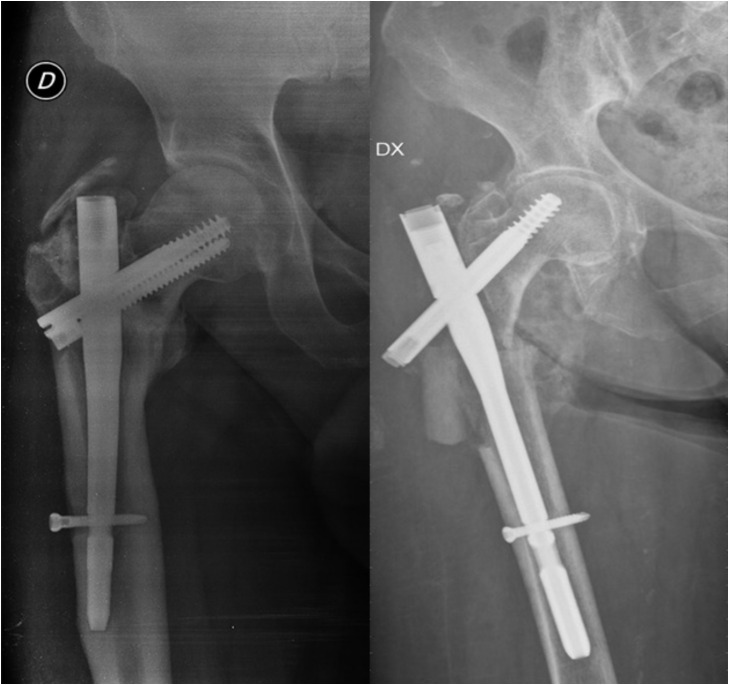
Failure examples of hip fracture fixation using nail devices.

**Figure 7 medicina-58-01314-f007:**
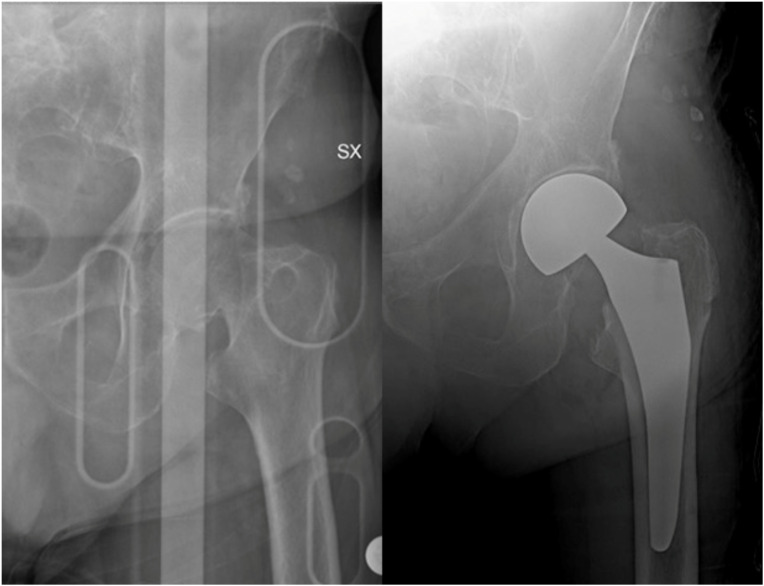
Treatment of fracture of neck of the femur using a hemiarthroplasty.

**Figure 8 medicina-58-01314-f008:**
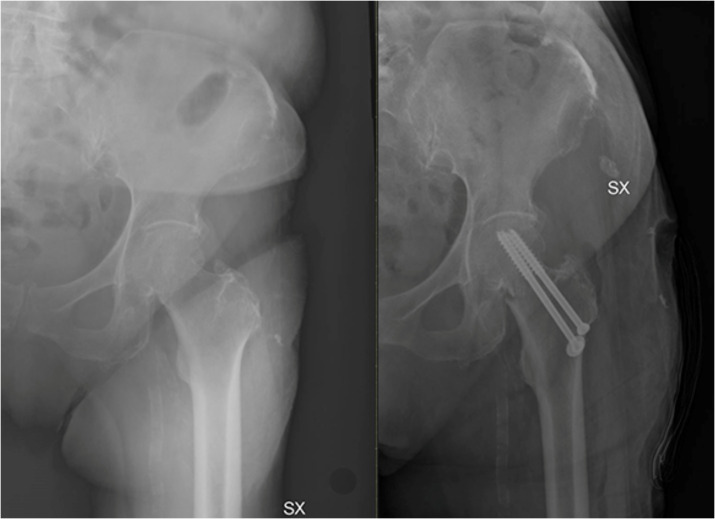
In young patients with non-displaced fractures or as a salvage option, two or three partially threaded canulated screws can be used.

**Figure 9 medicina-58-01314-f009:**
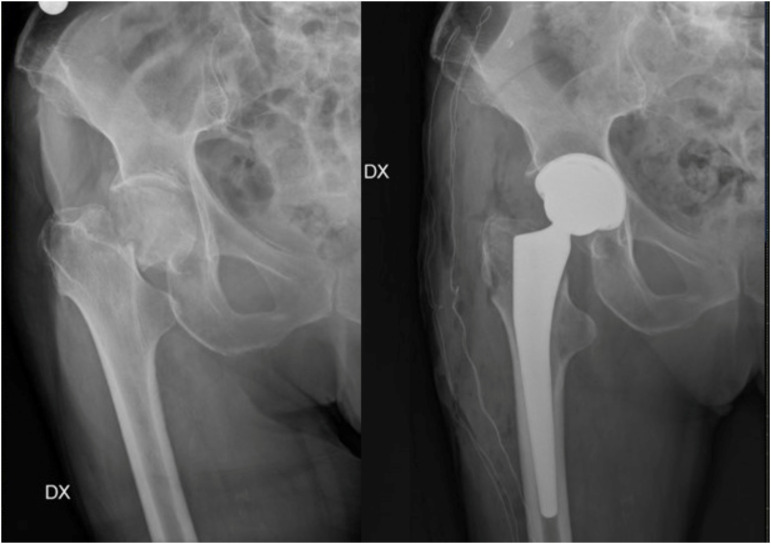
In case of healthy and active patients, with high functional requirements and lower biological age, total hip arthroplasty is the treatment of choice.

## Data Availability

Not applicable.

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
