# Peer review of "Proximal Femoral Fractures in the Elderly: A Few Things to Know, and Some to Forget"

_medicina, 2022, doi:10.3390/medicina58101314_

Round 1

Reviewer 1 Report

COMMENTS:

1.Paragraph 3-Classification. The AO classification should be better described in the test.

2. Paragraph 7 - Post-operative care: the statement "Secondary prevention of osteoporotic fracture with oral bisphosphonates is recommended in elderly with confirmed osteoporosis (T-score of −2.5 SD or below) "is questionable. In fact, the treatment decision depends on the risk of re-fracture and not necessarily on BMD values. Moreover, the treatment cannot be limited to oral bisphosphonates ;  in fact, in patients at  higher risk  may be necessary an initial treatment with anabolic drugs (teriparatide, abaloparatide, romosozumab) followed by anticatabolic drugs (oral or intravenous bisphosphonates or denosumab).

Author Response

Dear Sir.

Thank you for having allowed us to revise the above manuscript. The comments and suggestions of the reviewers have been carefully considered, and implemented as follows:

  1. Paragraph 3-Classification. The AO classification should be better described in the test.

Response: Thank you for this point, we considered the suggestion ad improved the AO classification description as follow from line 108 to line 114 “The original version (AO)/ASIF classification for hip fracture has been in use since 1990 [31], and has rapidly became popular and readily used in the scientific literature. The new AO/OTA classification, published in 2018, imparts greater importance to the integrity of the lateral wall, which may play an important role in decision making and has been identified as a major prognostic factor to predict mechanical failure after surgery [32]. Furthermore, the AO/OTA classification considers isolated trochanteric fractures (of the greater or lesser trochanter), which were not classified in the original AO system.”

  1. Paragraph 7 - Post-operative care: the statement "Secondary prevention of osteoporotic fracture with oral bisphosphonates is recommended in elderly with confirmed osteoporosis (T-score of −2.5 SD or below) "is questionable. In fact, the treatment decision depends on the risk of re-fracture and not necessarily on BMD values. Moreover, the treatment cannot be limited to oral bisphosphonates ;  in fact, in patients at  higher risk  may be necessary an initial treatment with anabolic drugs (teriparatide, abaloparatide, romosozumab) followed by anticatabolic drugs (oral or intravenous bisphosphonates or denosumab).

Response: Thank you for raising this issue. We re-wrote the paragraph following you suggestion from line 267 to line 271 “Secondary prevention of osteoporotic fracture is recommended in elderly patients with confirmed osteoporosis and high risk of re-fracture, with the initial use of anabolic drugs (such as teriparatide, abaloparatide, romosozumab) followed by anticatabolic drugs (i.e. oral or intravenous bisphosphonates or denosumab) [75]”

  1. As regards the introduction at line 48 i suggest improving this sentence citing the following articles: 

- The Prevalence of Fragility Fractures in a Population of a Region of Southern Italy Affected by Thyroid Disorders. Maccagnano G. et al. BioMed Research InternationalOpen AccessVolume 20162016 Article number 6017165

 -Epidemiology of diabetes mellitus in the fragility fracture population of a region of Southern Italy. Notarnicola A. et al. Journal of biological regulators and homeostatic agentsVolume 30, Issue 1, Pages 297 - 3021 January 2016

Response: Thank you, we cited these references.

  1. Furthermore, as regards the timing of treatment (from line 55 to 59) i suggest introducing at this part a sentence regarding the covid patients. In fact, according to literature there emerge an important discussion regarding first hit of femur fracture and second hit due to anaesthesia and citokin storm covid. I suggest citing these important references:

-SARS-CoV-2 infection and venous thromboembolism after surgery: an international prospective cohort study. Nepogodiev Dmitri et al. AnaesthesiaOpen AccessVolume 77, Issue 1, Pages 28 - 39January 2022.

-The epidemiology of proximal femur fractures during covid-19 emergency in italy: A multicentric study Ciatti C. et al. Acta BiomedicaVolume 92, Issue 53 November 2021 Article number e2021398.

-Direct Anterior versus Lateral Approach for Femoral Neck Fracture: Role in COVID-19 Disease. Maccagnano G. et al. J Clin Med 2022 Aug 16;11(16):4785.doi: 10.3390/jcm11164785.

Response: Thank you, we cited these references. We added from line 63 to line 68 the sentence “During the COVID-19 global pandemic, the total number of hip fracture patients was significantly reduced [18]. However, a systematic review and meta-analysis has shown a seven-fold increased mortality risk for COVID-19-positive patients with hip fractures and an increase in postoperative complications [19]. The time necessary to obtain the results of COVID-19 tests, the reduced operating capacity, and the shortage of hospital staff were identified as the major challenges. A recent multicenter study showed a mean delay of 2.4 days to surgery, with a minimum of 0 days and a maximum of 13 days [20].”

  1. As regards the surgical management, it is important to cite at line 172, the following article in which the Authors reported the results of hydroxyapatite screws in lateral femur fractures. 

-The effect of hydroxyapatite coated screw in the lateral fragility fractures of the femur. A prospective randomized clinical study. Pesce V. et al. Journal of biological regulators and homeostatic agents. Volume 28, Issue 1, Pages 125 - 1322014 Jan-Mar

Response: Thank you, we cited this reference. We added the sentence from line 182 to line 184 “A recent prospective randomised controlled trial in hip fracture patients showed that the use of nail and cephalic hydroxyapatite coated screws results in higher mechanical stability and improved implant osteointegration compared to standard nailing [52]”

We hope that changes have improved the manuscript, so that it will have now reached the standard necessary for formal acceptance by Medicina. We look forward to hearing from you.

Best regards,

Nicola Maffulli

Dear Sir.

Thank you for having allowed us to revise the above manuscript. The comments and suggestions of the reviewers have been carefully considered, and implemented as follows:

  1. Paragraph 3-Classification. The AO classification should be better described in the test.

Response: Thank you for this point, we considered the suggestion ad improved the AO classification description as follow from line 108 to line 114 “The original version (AO)/ASIF classification for hip fracture has been in use since 1990 [31], and has rapidly became popular and readily used in the scientific literature. The new AO/OTA classification, published in 2018, imparts greater importance to the integrity of the lateral wall, which may play an important role in decision making and has been identified as a major prognostic factor to predict mechanical failure after surgery [32]. Furthermore, the AO/OTA classification considers isolated trochanteric fractures (of the greater or lesser trochanter), which were not classified in the original AO system.”

  1. Paragraph 7 - Post-operative care: the statement "Secondary prevention of osteoporotic fracture with oral bisphosphonates is recommended in elderly with confirmed osteoporosis (T-score of −2.5 SD or below) "is questionable. In fact, the treatment decision depends on the risk of re-fracture and not necessarily on BMD values. Moreover, the treatment cannot be limited to oral bisphosphonates ;  in fact, in patients at  higher risk  may be necessary an initial treatment with anabolic drugs (teriparatide, abaloparatide, romosozumab) followed by anticatabolic drugs (oral or intravenous bisphosphonates or denosumab).

Response: Thank you for raising this issue. We re-wrote the paragraph following you suggestion from line 267 to line 271 “Secondary prevention of osteoporotic fracture is recommended in elderly patients with confirmed osteoporosis and high risk of re-fracture, with the initial use of anabolic drugs (such as teriparatide, abaloparatide, romosozumab) followed by anticatabolic drugs (i.e. oral or intravenous bisphosphonates or denosumab) [75]”

  1. As regards the introduction at line 48 i suggest improving this sentence citing the following articles: 

- The Prevalence of Fragility Fractures in a Population of a Region of Southern Italy Affected by Thyroid Disorders. Maccagnano G. et al. BioMed Research InternationalOpen AccessVolume 20162016 Article number 6017165

 -Epidemiology of diabetes mellitus in the fragility fracture population of a region of Southern Italy. Notarnicola A. et al. Journal of biological regulators and homeostatic agentsVolume 30, Issue 1, Pages 297 - 3021 January 2016

Response: Thank you, we cited these references.

  1. Furthermore, as regards the timing of treatment (from line 55 to 59) i suggest introducing at this part a sentence regarding the covid patients. In fact, according to literature there emerge an important discussion regarding first hit of femur fracture and second hit due to anaesthesia and citokin storm covid. I suggest citing these important references:

-SARS-CoV-2 infection and venous thromboembolism after surgery: an international prospective cohort study. Nepogodiev Dmitri et al. AnaesthesiaOpen AccessVolume 77, Issue 1, Pages 28 - 39January 2022.

-The epidemiology of proximal femur fractures during covid-19 emergency in italy: A multicentric study Ciatti C. et al. Acta BiomedicaVolume 92, Issue 53 November 2021 Article number e2021398.

-Direct Anterior versus Lateral Approach for Femoral Neck Fracture: Role in COVID-19 Disease. Maccagnano G. et al. J Clin Med 2022 Aug 16;11(16):4785.doi: 10.3390/jcm11164785.

Response: Thank you, we cited these references. We added from line 63 to line 68 the sentence “During the COVID-19 global pandemic, the total number of hip fracture patients was significantly reduced [18]. However, a systematic review and meta-analysis has shown a seven-fold increased mortality risk for COVID-19-positive patients with hip fractures and an increase in postoperative complications [19]. The time necessary to obtain the results of COVID-19 tests, the reduced operating capacity, and the shortage of hospital staff were identified as the major challenges. A recent multicenter study showed a mean delay of 2.4 days to surgery, with a minimum of 0 days and a maximum of 13 days [20].”

  1. As regards the surgical management, it is important to cite at line 172, the following article in which the Authors reported the results of hydroxyapatite screws in lateral femur fractures. 

-The effect of hydroxyapatite coated screw in the lateral fragility fractures of the femur. A prospective randomized clinical study. Pesce V. et al. Journal of biological regulators and homeostatic agents. Volume 28, Issue 1, Pages 125 - 1322014 Jan-Mar

Response: Thank you, we cited this reference. We added the sentence from line 182 to line 184 “A recent prospective randomised controlled trial in hip fracture patients showed that the use of nail and cephalic hydroxyapatite coated screws results in higher mechanical stability and improved implant osteointegration compared to standard nailing [52]”

We hope that changes have improved the manuscript, so that it will have now reached the standard necessary for formal acceptance by Medicina. We look forward to hearing from you.

Best regards,

Nicola Maffulli

Reviewer 2 Report

Dear Authors, notwithstanding the topic is discussed widly, your analysis is estremely interesting .

As regards the introduction at line 48 i suggest to improve this sentence citing the following articles: 

-The Prevalence of Fragility Fractures in a Population of a Region of Southern Italy Affected by Thyroid Disorders

Maccagnano G. et al.

BioMed Research InternationalOpen AccessVolume 20162016 Article number 6017165

-Epidemiology of diabetes mellitus in the fragility fracture population of a region of Southern Italy

Notarnicola A. et al.

Journal of biological regulators and homeostatic agentsVolume 30, Issue 1, Pages 297 - 3021 January 2016

In fact, it is important to underline that the epidemiological relationship between metabolics and inflammatory disease and fragility fractures is described already in literature  

Furthemore, as regards the timing of treatment ( from line 55 to 59) i suggest to introduce at this part a sentence regarding the covid patients. In fact according to literature there emerge an important discussion regarding first hit of femur fracture and second hit due to anaesthesia and citokin storm covid.

I suggest to cite these important references:

-SARS-CoV-2 infection and venous thromboembolism after surgery: an international prospective cohort study

Nepogodiev Dmitri et al.

AnaesthesiaOpen AccessVolume 77, Issue 1, Pages 28 - 39January 2022

-The epidemiology of proximal femur fractures during covid-19 emergency in italy: A multicentric study

Ciatti C. et al.

Acta BiomedicaVolume 92, Issue 53 November 2021 Article number e2021398

-Direct Anterior versus Lateral Approach for Femoral Neck Fracture: Role in COVID-19 Disease

Maccagnano G. et al.

J Clin Med 2022 Aug 16;11(16):4785.doi: 10.3390/jcm11164785

As regards the surgical management, it is important to cite at line 172, the following article in which the Authors reported the results of hydroxyapatite screws in lateral femur fractures. 

-The effect of hydroxyapatite coated screw in the lateral fragility fractures of the femur. A prospective randomized clinical study.

Pesce V. et al.

Journal of biological regulators and homeostatic agentsVolume 28, Issue 1, Pages 125 - 1322014 Jan-Mar

The augmentation technique (cemented spiral blade or  hydroxyapatite screws) is an important option to take into consideration in order to avoid the failure of implants  

As regards the conclusions, the section is balanced and well supported by analysis

Author Response

Dear Sir.

Thank you for having allowed us to revise the above manuscript. The comments and suggestions of the reviewers have been carefully considered, and implemented as follows:

  1. Paragraph 3-Classification. The AO classification should be better described in the test.

Response: Thank you for this point, we considered the suggestion ad improved the AO classification description as follow from line 108 to line 114 “The original version (AO)/ASIF classification for hip fracture has been in use since 1990 [31], and has rapidly became popular and readily used in the scientific literature. The new AO/OTA classification, published in 2018, imparts greater importance to the integrity of the lateral wall, which may play an important role in decision making and has been identified as a major prognostic factor to predict mechanical failure after surgery [32]. Furthermore, the AO/OTA classification considers isolated trochanteric fractures (of the greater or lesser trochanter), which were not classified in the original AO system.”

  1. Paragraph 7 - Post-operative care: the statement "Secondary prevention of osteoporotic fracture with oral bisphosphonates is recommended in elderly with confirmed osteoporosis (T-score of −2.5 SD or below) "is questionable. In fact, the treatment decision depends on the risk of re-fracture and not necessarily on BMD values. Moreover, the treatment cannot be limited to oral bisphosphonates ;  in fact, in patients at  higher risk  may be necessary an initial treatment with anabolic drugs (teriparatide, abaloparatide, romosozumab) followed by anticatabolic drugs (oral or intravenous bisphosphonates or denosumab).

Response: Thank you for raising this issue. We re-wrote the paragraph following you suggestion from line 267 to line 271 “Secondary prevention of osteoporotic fracture is recommended in elderly patients with confirmed osteoporosis and high risk of re-fracture, with the initial use of anabolic drugs (such as teriparatide, abaloparatide, romosozumab) followed by anticatabolic drugs (i.e. oral or intravenous bisphosphonates or denosumab) [75]”

  1. As regards the introduction at line 48 i suggest improving this sentence citing the following articles: 

- The Prevalence of Fragility Fractures in a Population of a Region of Southern Italy Affected by Thyroid Disorders. Maccagnano G. et al. BioMed Research InternationalOpen AccessVolume 20162016 Article number 6017165

 -Epidemiology of diabetes mellitus in the fragility fracture population of a region of Southern Italy. Notarnicola A. et al. Journal of biological regulators and homeostatic agentsVolume 30, Issue 1, Pages 297 - 3021 January 2016

Response: Thank you, we cited these references.

  1. Furthermore, as regards the timing of treatment (from line 55 to 59) i suggest introducing at this part a sentence regarding the covid patients. In fact, according to literature there emerge an important discussion regarding first hit of femur fracture and second hit due to anaesthesia and citokin storm covid. I suggest citing these important references:

-SARS-CoV-2 infection and venous thromboembolism after surgery: an international prospective cohort study. Nepogodiev Dmitri et al. AnaesthesiaOpen AccessVolume 77, Issue 1, Pages 28 - 39January 2022.

-The epidemiology of proximal femur fractures during covid-19 emergency in italy: A multicentric study Ciatti C. et al. Acta BiomedicaVolume 92, Issue 53 November 2021 Article number e2021398.

-Direct Anterior versus Lateral Approach for Femoral Neck Fracture: Role in COVID-19 Disease. Maccagnano G. et al. J Clin Med 2022 Aug 16;11(16):4785.doi: 10.3390/jcm11164785.

Response: Thank you, we cited these references. We added from line 63 to line 68 the sentence “During the COVID-19 global pandemic, the total number of hip fracture patients was significantly reduced [18]. However, a systematic review and meta-analysis has shown a seven-fold increased mortality risk for COVID-19-positive patients with hip fractures and an increase in postoperative complications [19]. The time necessary to obtain the results of COVID-19 tests, the reduced operating capacity, and the shortage of hospital staff were identified as the major challenges. A recent multicenter study showed a mean delay of 2.4 days to surgery, with a minimum of 0 days and a maximum of 13 days [20].”

  1. As regards the surgical management, it is important to cite at line 172, the following article in which the Authors reported the results of hydroxyapatite screws in lateral femur fractures. 

-The effect of hydroxyapatite coated screw in the lateral fragility fractures of the femur. A prospective randomized clinical study. Pesce V. et al. Journal of biological regulators and homeostatic agents. Volume 28, Issue 1, Pages 125 - 1322014 Jan-Mar

Response: Thank you, we cited this reference. We added the sentence from line 182 to line 184 “A recent prospective randomised controlled trial in hip fracture patients showed that the use of nail and cephalic hydroxyapatite coated screws results in higher mechanical stability and improved implant osteointegration compared to standard nailing [52]”

We hope that changes have improved the manuscript, so that it will have now reached the standard necessary for formal acceptance by Medicina. We look forward to hearing from you.

Best regards,

Nicola Maffulli